# Comparative Physiological and Transcriptome Analysis Provide Insights into the Response of *Cenococcum geophilum*, an Ectomycorrhizal Fungus to Cadmium Stress

**DOI:** 10.3390/jof8070724

**Published:** 2022-07-12

**Authors:** Yuyu Shi, Tianyi Yan, Chao Yuan, Chaofeng Li, Christopher Rensing, Yahua Chen, Rongzhang Xie, Taoxiang Zhang, Chunlan Lian

**Affiliations:** 1International Joint Laboratory of Forest Symbiology, College of Forestry, Fujian Agriculture and Forestry University, Fuzhou 350002, China; yuevan2014@163.com (Y.S.); m15246807099@163.com (T.Y.); yuanchao507@163.com (C.Y.); 2Asian Research Center for Bioresource and Environmental Sciences, School of Agricultural and Life Sciences, The University of Tokyo, 1-1-1 Midori-cho, Nishitokyo, Tokyo 188-0002, Japan; lichaofeng.1988cas@gmail.com (C.L.); lian@anesc.u-tokyo.ac.jp (C.L.); 3College of Resources and Environment, Fujian Agriculture and Forestry University, Fuzhou 350002, China; crensing94@gmail.com; 4College of Life Sciences, Nanjing Agricultural University, Nanjing 210095, China; yahuachen@njau.edu.cn; 5Forestry Bureau, Sanyuan District, Sanming 365000, China; xlz8365096@sina.com

**Keywords:** *Cenococcum geophilum*, mycorrhiza, cadmium, redox, membrane, ion transport

## Abstract

Cadmium (Cd) displays strong toxicity, high mobility, and cannot be degraded, which poses a serious threat to the environment. *Cenococcum geophilum* (*C. geophilum*) is one of the most common ectomycorrhizal fungi (ECMF) in the natural environment. In this study, three Cd sensitive and three Cd tolerant strains of *C. geophilum* were used to analyze the physiological and molecular responses to Cd exposure. The results showed that Cd inhibited the growth of all strains of *C. geophilum* but had a less toxic effect on the tolerant strains, which may be correlated to a lower content of Cd and higher activity of antioxidant enzymes in the mycelia of tolerant strains. Comparative transcriptomic analysis was used to identify differentially expressed genes (DEGs) of four selected *C. geophilum* strains after 2 mg/L Cd treatment. The results showed that the defense response of *C. geophilum* strain to Cd may be closely related to the differential expression of functional genes involved in cell membrane ion transport, macromolecular compound metabolism, and redox pathways. The results were further confirmed by RT-qPCR analysis. Collectively, this study provides useful information for elucidation of the Cd tolerance mechanism of ECMF.

## 1. Introduction

Cadmium (Cd) is one of the most toxic heavy metals [1,2], which can persist in soil for a long period of time [3]. It is estimated that approximately 25,000 tons of Cd is dumped into the environment every year through wastewater discharged from the chemical industry, combustion of municipal waste, and use of pesticides containing heavy metal, etc. [4,5]. Right now, the area of Cd contaminated soil has reached 2 × 10^5^ km^2^, thereby involving most provinces in China and causing serious harm to human health and ecology [6]. Cd not only causes environmental pollution but also destroys vegetation, reduces biomass, and disturbs the balance of the ecological system [7,8,9,10,11]. More seriously, Cd is also detrimental to human health [12]. Therefore, the restoration of heavy metal contaminated soil is critical for the protection of human and ecological security.

Fungi and plants are important components of the natural ecosystem [13]. More than 90% of vascular plants have a symbiotic relationship with mycorrhiza fungi [14]. In this symbiotic relationship, fungi provide water and mineral elements such as nitrogen and phosphorus for plants, while plants provide carbohydrate for fungi [15]. Based on the morphological structure of mycorrhizal fungi invading plant roots, mycorrhizal fungi can be divided into four main types: arbuscular mycorrhiza (AM), ectomycorrhizal fungi (ECMF), orchid mycorrhiza (OM), and ericoid mycorrhizas (ERM) [16]. There are approximately 5000–6000 species of ECMF in the world [17]. ECMF have been shown to expand the absorption area of host plant roots, enhance nutrient absorption, promote plant growth, and regulate antioxidant defense systems to improve the disease and stress resistance of host plants [18,19,20,21]. A large number of studies have shown that ECMF are widely distributed in heavy metal polluted areas and are able to improve the resistance of host plants to heavy metals [22]. Hence, ECMF have good prospects for application in vegetation restoration of heavy metal contaminated soils.

ECMF can not only reduce the absorption of excessive heavy metals by plants, but also absorb and store heavy metals by themselves so as to improve the ability of host plants to resist the toxicity of heavy metals [23,24,25]. The metabolites of ECMF (such as iron carrier, succinic acid, formic acid, oxalic acid, etc.) are able to adjust the availability of soil metal ions and reduce the toxicity of heavy metals [26,27]. Luo et al., reported that the chitin, melanin, cellulose, and their derivatives in fungal cells can provide binding sites for heavy metals, and the heavy metals ions will be immobilized or restricted in the fungal cell wall to reduce the toxicity of heavy metals [28]. According to electron energy loss spectroscopy and electron spectral images, heavy metals in *Pisolithus arrhizus* mycelium are mainly located in pigments and electron opaque particles deposited on the surface of the fungal cell wall [29]. *Tricholoma* bicolor secreted polyphosphates to chelate heavy metal ions in vacuoles, thereby isolating them from other components in the cell and reduce the toxicity of heavy metals in the cytoplasm [30]. ECMF can increase the activity of antioxidant enzyme under heavy metal stress, thereby reducing the damage caused by oxidative stress. Yin et al. showed that the Cd and cuprum (Cu) treatments caused a significant increase in the antioxidant enzyme activities of the *Lepista sordida* [31]. Moreover, ECMF can modulate the physiological mechanism by changing the expression of genes so as to resist heavy metal. In the presence of high concentrations of nickel, the genes encoding p-type atp enzyme, ABC transporter, and the major facilitator superfamily permease (MFS) play an important role in achieving nickel tolerance [32]. The genes in *Trichoderma harzianum* encoding functions in cellular homeostasis, transcription initiation, sulfur compound biosynthesis, metabolic processes, RNA processing, protein modification, and vesicle-mediated transport were up-regulated under elevated Cd concentrations, while genes encoding functions in carbohydrate metabolism were down-regulated [33]. These reports indicate that ECMF have specific ecological adaptability and response mechanism to heavy metal treatment.

*Cenococcum geophilum Fr.* (*C. geophilum*), which belongs to the phylum Ascomycota, is one of the most common ECMF in the natural environment. *C. geophilum* cannot produce fruit bodies, and generally exists in structures of mycelium, sclerotium, and mycorrhiza [34,35,36,37]. *C. geophilum* has a rich host plant and is widely distributed in a variety of harsh environments, including in highly polluted sites, arid, and saline environments, which are not conducive to the growth of mycorrhizal fungi [38,39]. Studies have shown that *C. geophilum* isolated from different geographical locations and hosts maintains a high degree of genetic diversity [40,41,42]. The inoculation of *C. geophilum* promoted the growth of host plant roots, improved the absorption and utilization of mineral elements, and regulated the secretion of secondary metabolites of the host plants, thereby enhancing the ability of plants to resist abiotic and biotic stress [43,44,45]. Numerous studies have shown that *C. geophilum* was able to improve the survival of host plants by increasing the biomass, photosynthesis, and mineral nutrient absorption, and weakened the transfer of Cd, lead (Pb), Cu, and zinc (Zn) from soil to plant roots [46,47]. At present, there are many studies looking at the genetic diversity of *C. geophilum* and improving the tolerance of plants to heavy metals. However, few studies have focused on the response of *C. geophilum* itself to Cd stress. The mechanism of *C. geophilum* tolerance to Cd remains unclear, and there are few studies focused on the physiological and transcriptome changes of different strains of the same ECMF under Cd treatment.

In the present study, we selected six *C. geophilum* strains (from China and Japan) with contrasting Cd tolerance to investigate the difference in enzyme activity, nutrient element content, Cd distribution, and gene expression between sensitive and tolerant strains under Cd treatment to further explore the response mechanism of *C. geophilum* to Cd stress. The results of this research are expected to provide the scientific basis for ecological reconstruction and phytoremediation in heavy metal polluted areas.

## 2. Materials and Methods

### 2.1. Mycelial Growth of C. geophilum Strains under Different Cd Treatments

Three sensitive strains (C1, J37, and J45) and three tolerant strains (C78, J127, and J202) evaluated in previous experiments of the International Joint Laboratory of Forest Symbiology at Fujian Agriculture and Forestry University were used in this study (Appendix A).

The six strains were pre-cultured in ordinary modified melin-norkrans (MMN) agar medium for 45 days in the dark at 23 °C, then a viable agar plug (7 mm in diameter) was taken into the center of each petri dish (diameter 9 cm) containing 20 mL MMN agar medium for different Cd concentration treatments (0, 0.25, and 2 mg/L Cd) and cultured for 30 days in the dark at 23 °C. Three replicates of each strain were conducted for each treatment. After culturing for 30 days, the mycelial area of each strain was measured by X-Plan 380dⅢ, Ushikata (Kantum Ushikata Co., Ltd., Yokohama, Japan).

### 2.2. Determination of Superoxide Dismutase (SOD), Peroxidase (POD), and Catalase (CAT) in Mycelium

The mycelia of six strains were pre-cultured for 30 days in the dark at 23 °C in MMN agar medium covered with a layer of cellophane. The pre-cultured mycelia were transferred into a triangular flask containing 50 mL of liquid MMN culture medium and cultured for 30 days, then followed by Cd treatment (0 and 2 mg/L Cd) for 15 days. Mycelia were collected and washed three times with phosphate buffer saline (PBS) to remove culture medium, and then divided into two parts. One was stored at −80 °C for determining antioxidant enzyme activity, and the other was dried at 80 °C for 48 h for determining the contents of nutrient elements and heavy metals. A reagent test kit (Nanjing Jiancheng Bioengineering Institute, Nanjing, China) was used to determine the activities of SOD, POD, and CAT in the mycelia of *C. geophilum*.

### 2.3. Determination of Nutrient Element and Heavy Metal Contents in Mycelia

The dried mycelia were digested with HNO_3_ and H_2_O_2_ (V:V = 5:1) in an automatic digestion instrument for 1.5 h. Ultrapure water was added to dilute the digestion solution up to 25 mL in a volumetric flask, and then filtered with qualitative filter paper and 0.45 µm filter membrane successively. Finally, the inductively coupled plasma optical emission spectrometry (ICP-OES, PEOPTIMA8000, PerkinElmer, Chiba, Japan) was used to determine the contents of phosphorus (P), kalium (K), natrium (Na), calcium (Ca), magnesium (Mg), ferrum (Fe), aluminum (Al), manganese (Mn), Cu, Cd, and Zn.

### 2.4. Structure and Cd Distribution of Cells in Mycelia

Two strains, J45 (sensitive) and J127 (tolerant), were selected to analyze the distribution of Cd in the mycelium. The mycelia of two strains were pre-cultured for 30 days in the dark at 23 °C in MMN agar medium covered with a layer of cellophane. The mycelia of pre-cultured J45 and J127 were transferred into a triangular flask containing 50 mL of liquid MMN culture medium and cultured for 30 days, followed by Cd treatment (0 and 2 mg/L Cd) for 15 days. Mycelia were washed twice with PBS buffer (10 mM, pH 7.2), then suspended in 2.5% glutaraldehyde fixative solution, and stored at 4 °C for fixed preservation. A scanning electron microscopy coupled with energy dispersive X-ray (SEM-EDX; EDAX Inc. Genesis XM, Mawah, NJ, USA) analysis was performed to observe the immobilization of Cd in the hyphae of J45 and J127.

### 2.5. RNA-Seq Analysis

The four selective strains (C1, J45, C78, and J127) were pre-cultured for 30 days in the dark at 23 °C in MMN agar medium covered with a layer of cellophane. The pre-cultured mycelia were transferred into a triangular flask containing 50 mL of liquid MMN culture medium and cultured at 23 °C for 30 days, and then 50 µL of 2 g/L Cd was added to each flask (final concentration of Cd: 2 mg/L) and further cultured at 23 °C for 24 h. For the control (0 mg/L Cd), 50 µL of sterilized water was added. After culturing, the mycelia were collected with a sieve, washed three times with PBS buffer to remove the medium, and the mycelia were stored in a −80 °C refrigerator.

The extraction of RNA from the *C. geophilum* mycelium was conducted by the Trizol method. A total of 1.5 µg of RNA per sample was used for the construction of cDNA library. Construction of the cDNA library, quality verification, and further sequencing were performed by Biomarker Technologies (Beijing, China) using an Illumina NovaSeq 6000 platform in accordance with standard protocols. The clean reads were mapped to the *C. geophilum* strain 1.58 reference genome v2.0 (Cenge3, https://mycocosm.jgi.doe.gov/Cenge3/Cenge3.home.html, accessed on 1 December 2021) using the HISAT2 software with default parameters [48], then the reads were assembled into transcripts to compare with reference genes using the StringTie software [49]. Differentially expressed genes (DEGs) of mycelia between the cadmium treatment and control were identified and annotated using the DESeq2 tool [50]. |log2Foldchange| ≥ 1.5 and *p*-value < 0.05 were used as the screening criteria for DEGs. The novel genes identified in the enrichment analysis were ruled out in the further analysis. Further Kyoto Encyclopedia of Genes and Genomes (KEGG) and Gene Ontology Consortium (GO) analysis were used to annotate and conduct the DEGs with the KEGG database [51] and PlantGSEA software [52], respectively.

### 2.6. RT-qPCR Analysis

Combined with the transcriptome analysis, physiological indices, and enzyme activity analysis, 10 DEGs of *C. geophilum* were selected for RT-qPCR analysis. The *C. geophilum* 18S rRNA (LoBuglio, Berbee et al., 1996) was used as the reference gene. The tool Primer Premier 5.0 (http://www.premierbiosoft.com/, accessed on 1 February 2022) was used to design the primers for the candidate genes. Primers sequences for RT-qPCR were listed in Appendix A. Step One Real-Time PCR fluorescence quantitative PCR instrument was used to detect the Ct value of each template, and the relative expression changes of the target gene in the control group and the experimental group were calculated by 2^−ΔΔCt^ method [53].

### 2.7. Statistical Analysis

The significant differences of mycelial growth for each strain among the different Cd treatments (0, 0.25, and 2 mg/L Cd) were tested by using one-way analysis of variance (ANOVA) followed by Duncan’s test at a level of *p* < 0.05 using SPSS ver. 21.0, and the R package was used for Perform difference analysis. All the assays were performed with three replicates for each treatment. Results are presented as the mean value and standard deviation.

## 3. Results

### 3.1. Effects of Cd Treatment on C. geophilum Mycelial Growth

The relative growth of six *C. geophilum* mycelia treated by Cd (0, 0.25, and 2 mg/L Cd) was used to evaluate the treatment response of *C. geophilum* to Cd. The results showed that Cd treatments seriously affected the growth of the sensitive strains (C1, J37, and J45) (Figure 1). Under 2 mg/L Cd treatment, the mycelia of J37 and J45 did not grow and the relative growth of C1 was 1.15% compared to the control (*p* < 0.001). For C78, J127, and J202 (tolerant strains), although the mycelial growth was also significantly inhibited by the 2 mg/L Cd treatment (*p* < 0.001), the relative growth of C78, J127, and J202 was relatively high and reached 67.50%, 80.99%, and 55.54%, respectively (Figure 1).

### 3.2. The Activities of SOD, POD, and CAT

Effects of Cd treatment on the antioxidant enzymes of *C. geophilum* are shown in Figure 2. Under 2 mg/L Cd treatment, the activities of SOD, POD, and CAT in all six Cg strains were significantly higher than those of the control (*p* < 0.05). Compared between sensitive and tolerance strains, the SOD and CAT contents of tolerant strains were higher than those of sensitive strains (Figure 2). However, no significant difference in POD activity was observed between sensitive and tolerance strains under Cd treatment (Figure 2b).

### 3.3. The Contents of Nutrient Elements and Cd

Cd treatment of 2 mg/L Cd increased the contents of P, K, Na, Ca, Mg, Fe, Al, Mn, Cu, and Zn of mycelia of all the strains compared to the control (0 mg/L Cd treatment), but there was no significant difference in the nutrient element contents of mycelia between the sensitive (C1, J37, and J45) and tolerant (C78, J127, and J202) strains (Appendix A).

For Cd content, Cd was not detected in mycelia of any strain of *C. geophilum* in the control treatment. The supply of 2 mg/L Cd significantly increased the Cd content in mycelia of all strains of *C. geophilum* (*p* < 0.05) (Figure 3). The Cd contents of the three sensitive strains (C1, J37, and J45) ranged between 0.4 and 0.8 mg/Kg. For the tolerant strains, except for J202 (Cd content, 0.65 mg/Kg), the Cd contents of C78 and J127 were 0.2 and 0.4 mg/Kg, which was significantly lower than those in mycelia of the sensitive strains (Figure 3).

### 3.4. Cd Distribution in Cells of J45 and J127

The transmission electron microscopy (TEM) was conducted to analyze the distribution of Cd in the cell of J45 (sensitive strain) and J127 (tolerant strain) (Figure 4).

Under the 0 mg/L Cd treatment, both J127 and J45 had round and full cell morphology, and the cell wall was neat and smooth (Figure 4a,e). Under the concentration of 2 mg/L Cd, the cell structures of J127 and J45 showed different changes (Figure 4b–d,f–h). The cells of both strains became smaller, and the wall become thicker (the blue arrow points to the part) compared to those in the treatment of 0 mg/L Cd. The cells of tolerant strain J127 shrank more than those of the sensitive strain J45. Moreover, there was a large amount of black matter accumulation at the edge of the cell wall of J127 (the purple arrow points to the part), and black shadows also presented in the cytoplasm and vacuoles (the red arrow points to the part) in both J127 and J45 after the Cd treatment. We speculate that it is the Cd or Cd complex entering the cell. However, the black shadow area in sensitive strain was larger than that of tolerant strain, indicating that the tolerant strain J127 may accumulate Cd outside the cell, while the sensitive strain J45 may accumulate Cd mainly inside the cell.

### 3.5. Cd Treatment Influences the Expression of Some DEGs

Combining the determination of physiological indicators and response patterns of *C. geophilum* strains to Cd treatment, two Cd-sensitive strains (C1 and J45), and two Cd-tolerant strains (C78 and J127) were randomly selected for RNA-seq analysis. From the RNA-seq analysis, about 120 Gb reads of all samples were obtained, and then mapped to reference Cenge3 database with nearly 75% mapping rate (Appendix A). More than ten thousand unigenes were expressed in each sample. Compared with their control groups, the number of DEGs for each *C. geophilum* strain was 1099 (C1), 815 (J45), 637 (J127), and 228 (C78) after 2 mg/L Cd treatment (Figure 5). The relative expression levels of these DEGs in the control and experimental groups are also shown in Appendix A.

Go enrichment analysis showed that some DEGs were enriched in different pathways of molecular function, cellular component, and biological processes (Figure 6). The Cd tolerant group shared the processes of “iron ion binding”, “cytoplasm”, and “oxidation-reduction process” (Figure 6b,d), while two Cd sensitive *C. geophilum* strains shared little process after 2 mg/L Cd treatment (Figure 6a,c). Some DEGs were also classified regulation of transcription, protein phosphorylation, and translation function (Figure 6). It is worth mentioning that many DEGs associated with ion transport were found in most of the *C. geophilum* strains.

To clarify the commonality and specificity of these DEGs from different *C. geophilum* strains, the Cd sensitive group (C1 and J45) and Cd tolerant group (J127 and C78) were used to draw a Venn diagram (Figure 7), respectively. From the Figure 7, we could find that most DEGs belonged to the unique genes for each *C. geophilum* strains. The number of unique genes were 922 (C1), 638 (J45), 623 (J127), and 214 (C78). There were also some shared genes for the different *C. geophilum* strains responses to Cd treatment. The Cd sensitive group shared 177 DEGs (Figure 7), while the Cd tolerant group only shared 14 DEGs (Figure 7). These DEGs showed various expression patterns after 2 mg/L Cd treatment (Figure 7 and Appendix A).

Hence, the KEGG enrichment analysis was used to identify the regulatory pathways of these DEGs when treated with Cd. The C1 strain shared the pathways of nitrogen metabolism, valine, leucine and isoleucine biosynthesis, alanine, aspartate, and glutamate metabolism, and 2-oxocarboxylic acid metabolism with J45 strain (Figure 8), whereas the DEGs of methane metabolism were enriched in the Cd-tolerant group after 2 mg/L Cd treatment (Appendix A). It is worth mentioning that many DEGs associated with secondary metabolism, such as fatty acid metabolism and glutathione metabolism, were also enriched (Figure 8 and Appendix A). For the enriched 4 common pathways of C1 and J45, 23 of DEGs were identified (Appendix A). Among them, some DEGs were all up-regulated after Cd treatment (gene-K441DRAFT_609805 and gene-K441DRAFT_672999), some DEGs were all down-regulated after Cd treatment (gene-K441DRAFT_611738, gene-K441DRAFT_653815 and gene-K441DRAFT_679972), while some DEGs (gene-K441DRAFT_571214 and gene-K441DRAFT_668338) had different expression patterns (Appendix A). We also found that 5 of DEGs were found in the common pathway of methane metabolism, and they had different expression patterns in the J127 and C78 strains (Appendix A).

### 3.6. RT-qPCR Verification of DEGs

According to the results of RNA-seq and expression patterns of *C. geophilum* to Cd treatment, ten eligible DEGs were selected for RT-qPCR analysis (Appendix A). The relative expression levels of these genes were presented in the Figure 9.

RT-qPCR analysis showed that five gene gene-K441DRAFT-552345, gene-K441DRAFT-600384, gene-K441DRAFT-611738, gene-K441DRAFT-660325, and gene-K441DRAFT-681328 were significantly down-regulated after Cd treatment (Figure 9a–e), while the other four genes (gene-K441DRAFT-653729, gene-K441DRAFT-654069, gene-K441DRAFT-668187, and gene-K441DRAFT-672999) were significantly up-regulated (Figure 9g–i), which are similar to the results of RNA-seq analysis. The results of RT-qPCR and RNA-seq analysis identified some candidate genes involved in the process of Cd treatment response.

## 4. Discussion

*Cenococcum geophilum* (*C. geophilum*) is one of the most common ECMF with a large variety of hosts and occupying a wide ecological niche, from alpine to tropical regions [54,55]. Compared to other ECMF, *C. geophilum* has a stronger ability to adapt to the geographical environment in a harsh habitat [38,39,56,57,58,59]. Dauphin et al., (2021) reported that *C. geophilum* from different geographical locations and different hosts have a rich genetic diversity and a different ability to tolerate heavy metals [60]. In this research, the physiological and transcriptional responses of sensitive and tolerant *C. geophilum* strains to Cd were compared. We found that under exposure to Cd, the activities of antioxidant enzymes of all the strains were increased (Figure 2). Moreover, transcriptome analysis showed that most of the DEGs are predicted to regulate oxidative processes (oxidative detoxification and oxidative stress), ion transmembrane transport, and macromolecular complex metabolism (methane metabolism, starch and sucrose metabolism, carbon metabolism and glyceride metabolism, etc.) pathways.

Heavy metal stress has been shown to cause ECMF to generate excessive amounts of reactive oxygen (such as superoxide radicals, hydroxyl radicals, hydrogen peroxide, etc.), which has strong chemical reactivity and attacks biological macromolecules resulting in oxidative damage [61]. ECMF harbor genes encoding various protective mechanisms to eliminate and reduce reactive oxygen free radicals (ROS), one of which is the antioxidant enzyme system, so as to mitigate the damage of H_2_O_2_ and lipid peroxidation [62]. Antioxidant enzymes (SOD, CAT, and POD) are able to effectively scavenge free radicals and reduce the concentration of reactive oxygen species to alleviate the physiological toxicity of heavy metals to cells. In this study, the activities of CAT, POD, and SOD in the six Cg mycelia were increased under Cd treatment, and the abundance of SOD and CAT in the tolerant strains were higher than those in the sensitive strains (Figure 2), which was consistent with previous research [63]. Lepista sordida displayed a significant increase in activity of antioxidant enzymes (SOD, POD, and CAT) under Cd and Cu treatment [31]. Moreover, in our research, we also observed that hundreds of genes encoding functions in oxidation-reduction processes, as a response to oxidative stress and cellular oxidant detoxification, were up-regulated in tolerant strains compared to sensitive strains under Cd treatment, indicating that Cd may stimulate the expression of genes involved in redox of the strains to maintain ROS balance (Figure 6 and Figure 8). In this regard, there have been few studies on ECMF, and the mechanism is yet unclear, but there have been studies on AM. Several EST sequences were found in the hyphae of *Rhizophagus irregularis* column, which may be involved in encoding enzyme responsible for scavenging ROS, such as glutathione (GST), superoxide dismutase activating enzyme (SOD), cytochrome P450, and Thioredoxin, etc. [64]. This indicates that the Cd resistance mechanism of ECMF is similar to that of AM. Lanfranco et al. identified a gene encoding Cu/Zn SOD GmarCuZn SOD in Gigaspora margarita, and the expression of this gene was able to reduce the damage by ROS caused by Phosphate (Na_2_HPO_4_·12H_2_O) to avoid oxidative stress [65]. Overall, oxidative stress defense and related antioxidant reductase played important roles in Cd resistance of *C. geophilum*.

TEM showed that tolerant strain of *C. geophilum* accumulated more extracellular Cd, while the sensitive strain accumulated more intracellular Cd (Figure 4).Therefore, we speculate that *C. geophilum* mainly mitigates the toxicity of Cd by preventing heavy metals ions from entering the cell. Studies have shown that the membrane can transport heavy metal ions out of the cell through ion exchange on the protoplast membrane [66]. Furthermore, heavy metal ions can be transported across the vacuolar membrane into the vacuole for compartmentalization, which is an important mechanism for EMF to tolerate heavy metals [67,68,69]. Through the whole-cell patch-clamp technique, Perfus-Barbeoch et al. confirmed that Cd^2+^ was able to permeate into roots through the plasma membrane (PM) Ca^2+^ channels in guard cells [70]. Ions channels that are permeable to Cd^2+^ in the PM of the fungi contributed to Cd^2+^ uptake [71], which may be the reason for the high Cd content in sensitive strains of *C. geophilum*. Through transcriptome analysis, genes encoding Cd tolerance determinants in cotton were mainly involved in metal binding processes, and the protein ghhmad5 containing a heavy metal binding domain was identified in the transport or detoxification pathway of heavy metal ions [72]. In this study, transcriptome results showed that *C. geophilum* was able to up-regulate genes encoding integral components of the membrane, integral components of the plasma membrane, membrane par, fungal-type vacuolar membranes, and other functions (Figure 6 and Figure 8), which was similar with some other studies [73]. Therefore, combined with the concentrations of Cd in sensitive and tolerant *C. geophilum* strains, the distribution of Cd in cells, and DEGs expression, we speculate that *C. geophilum* may bind and transport Cd through ion transport on the cell membrane to reduce the content of Cd entering the cell and to reduce the toxic effect of heavy metals.

Venn diagrams, GO, and KEGG enrichment analysis showed that most of the DEGs are unique genes for each *C. geophilum* strain (Figure 6, Figure 7 and Figure 8), which is consistent with the genetic diversity and variations in heavy mental resistance of *C. geophilum* [74]. The results also indicate that different *C. geophilum* strains may have produced some unique mechanisms for co-evolution with their various host plants [15]. Compared with the Cd sensitive strains, more regulatory pathways mediated the Cd treatment response in the Cd tolerant strains (Figure 6 and Figure 8). Meanwhile, calmodulin binding, metal ion binding, fungal-type cell wall, ATP biosynthetic process, heme binding, and membranes can be found in most *C. geophilum* strains after Cd treatment (Figure 6). All of these regulatory pathways can be found in the arbuscular mycorrhizal fungi under heavy metal treatment [75,76], and show that some mechanisms are conserved in the heavy metal response, especially Cd treatment. Moreover, DEGs of nitrogen metabolism, valine, leucine and isoleucine biosynthesis, alanine, aspartate and glutamate metabolism, and 2-oxocarboxylic acid metabolism, methane metabolism, and oxidative phosphorylation pathways were indicated to play key roles in the Cd-induced ROS homeostasis (Figure 8), which are inconsistent with high plants [77,78].

## 5. Conclusions

In this study, three Cd sensitive and three Cd tolerant strains of *C. geophilum* were selected as materials, and the responses to Cd were analyzed from the aspects of growth, enzyme activity, element absorption, and gene expression. The results showed that under Cd stress, *C. geophilum* could resist the toxicity of Cd to the strains by increasing the antioxidant enzyme activity and nutrient absorption, reducing the absorption of Cd, and regulating the related genes encoding metabolism of compounds, ion transport over the cell membrane and antioxidant enzyme activity. In summary, these results provide some new insights into the physiological and molecular responses of *C. geophilum* to Cd treatment, which may help researchers understand the Cd stress responses of ECMF.

## Figures and Tables

**Figure 1 jof-08-00724-f001:**
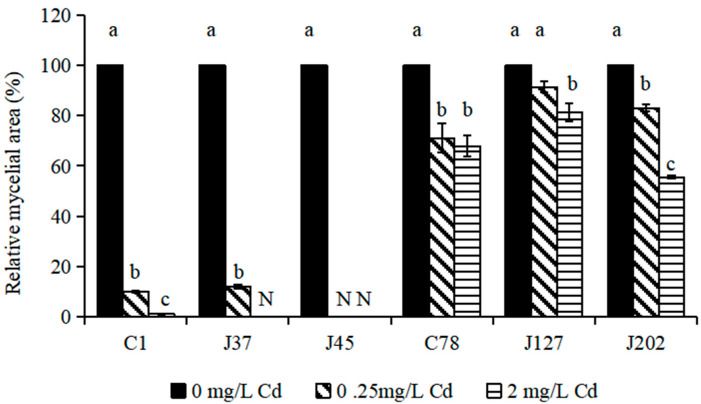
The relative growth of six strains of *C. geophilum* under different Cd concentrations. Relative mycelial area was calculated as ((mycelial area at a certain Cd concentration)/(mycelial area at 0 mg/L Cd)) × 100(%). Data were shown as mean ± SD of replicates (*n* = 3). Values with different letters are significantly different (Kruskal–Wallis test, α < 0.05). N represents that the isolate did not grow.

**Figure 2 jof-08-00724-f002:**
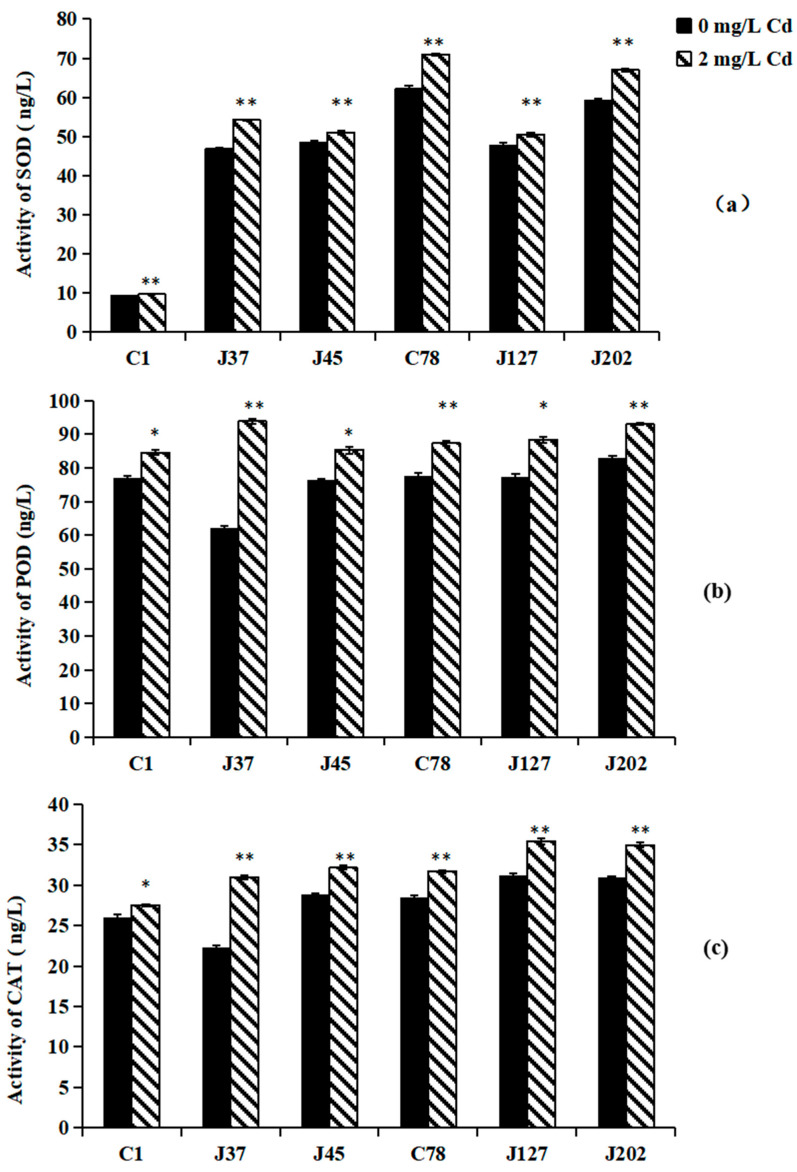
The activities of SOD (**a**), POD (**b**), and CAT (**c**) in the mycelia of six *C. geophilum* strains. Data were shown mean ± SD of replicates (*n* = 3). Significant difference of the activities of SOD, POD, and CAT after Cd treatment was analyzed by the student’s *t*-test. * *p* < 0.05; ** *p* < 0.001.

**Figure 3 jof-08-00724-f003:**
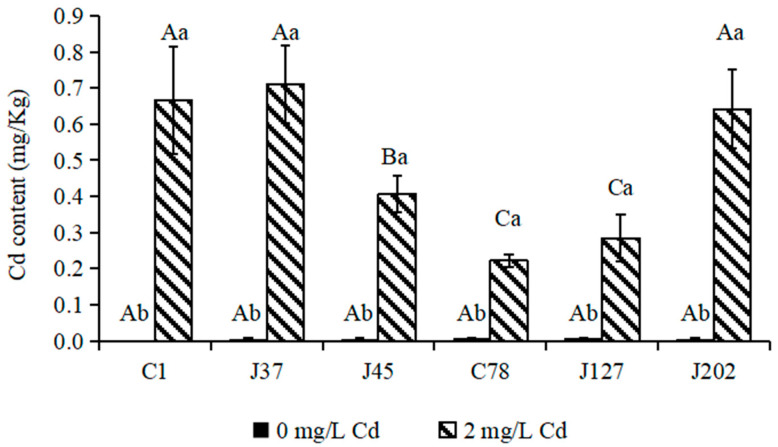
The Cd contents of mycelia of six *C. geophilum* strains. Uppercase letters indicate between-group differences, lowercase letters indicate within-group differences. Data were shown mean ± SD of replicates (*n* = 3). Significant difference of the content of each element of each isolate between 0 mg/L and 2 mg/L Cd concentrations was tested by the Welch t-test (* *p* < 0.05; ** *p* < 0.001).

**Figure 4 jof-08-00724-f004:**
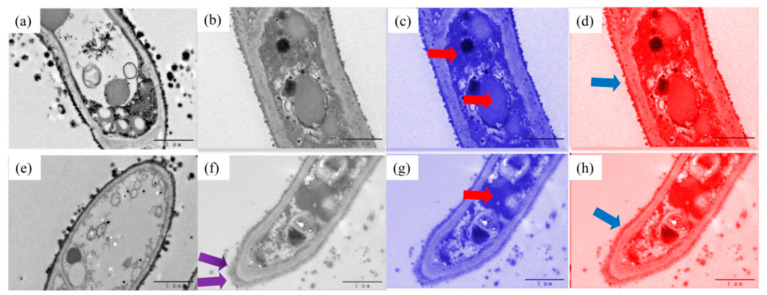
The transmission electron micrograph (TEM) images of mycelia of J45 (**a**–**d**) and J127 (**e**–**h**) under 0 and 2 mg/L Cd concentrations (resolution 1 µm), (**a**) J45 at 0 mg/L Cd concentration, (**b**–**d**) J45 at 2 mg/L Cd concentration, (**e**) J127 at 0 mg/L Cd concentration, (**f**–**h**) J127 at 2 mg/L Cd concentration, The blue arrow indicates that the cell wall is thickened, and the red arrow indicates the black shadows also presented in the cytoplasm and vacuoles, the purple arrow indicates the black matter accumulation at the edge of the cell wall.

**Figure 5 jof-08-00724-f005:**
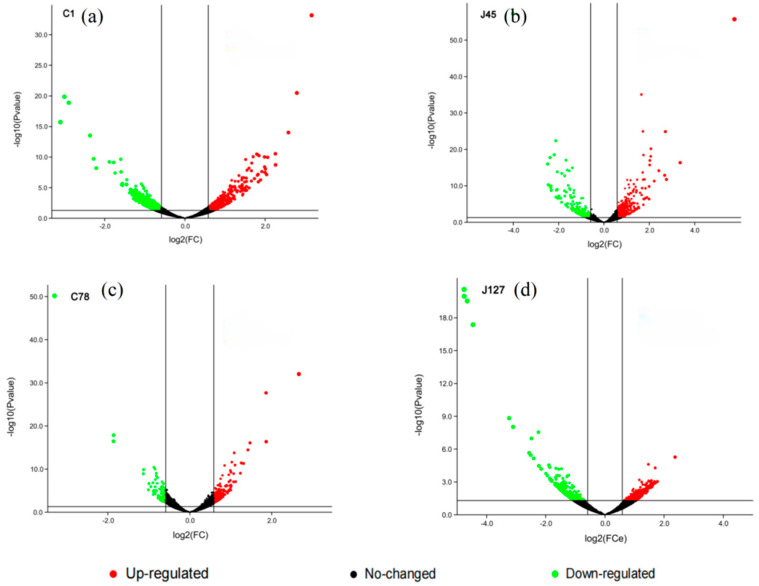
Volcano map of candidate genes induced by 2 mg/L Cd. Volcano map shows the DEGs in four selected *C. geophilum* strains after 2 mg/L Cd treatment as compared to the same strains cultured in normal conditions (0 mg/L Cd). Red spot indicates significant up-regulated genes after cadmium treatment; green spot indicates significantly down-regulated genes; black spot indicates no-change genes. |log2Foldchange| ≥ 2 was used as the screening criteria of DEGs.

**Figure 6 jof-08-00724-f006:**
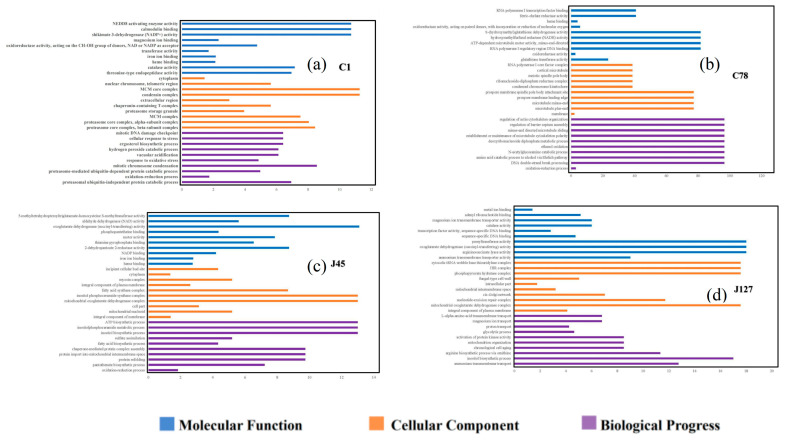
GO enrichment analysis of the DEGs in four strains of *C. geophilum* after 2 mg/L Cd treatment as compared to the same strains cultured in normal conditions (0 mg/L Cd). Y axis represents different regulatory pathway in the molecular function (blue), cellular component (yellow), and biological progress (purple) categories, respectively. *X*-axis represents the numbers of DEGs involved in the corresponding pathways.

**Figure 7 jof-08-00724-f007:**
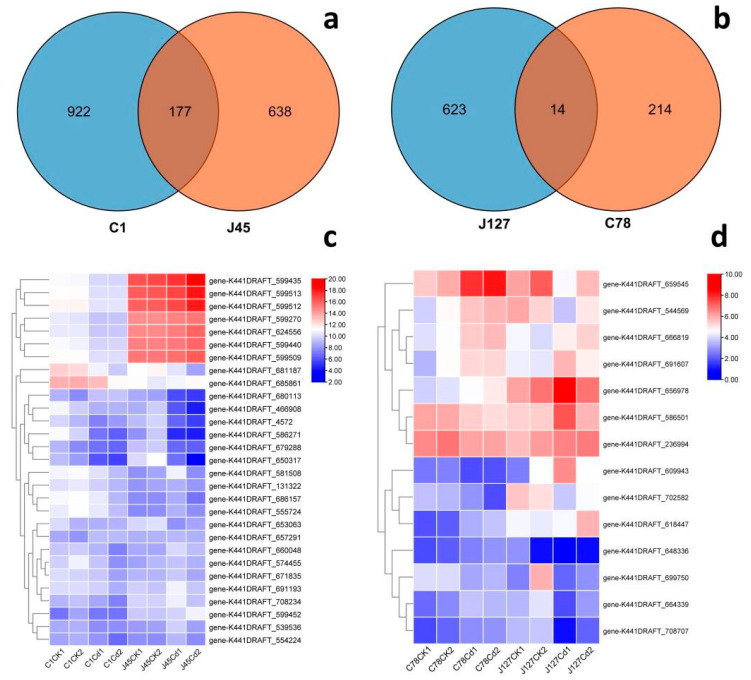
Genome-wide analysis of the DEGs in four strains of *C. geophilum* after 2 mg/L Cd treatment. (**a**) Venn diagram of total DEGs in two sensitive strains (C1 and J45) of *C. geophilum* after 2 mg/L Cd treatment as compared to the same strains cultured in normal conditions (0 mg/L Cd). (**b**) Venn diagram of total DEGs in two resistant strains (C78 and J127) of *C. geophilum* after 2 mg/L Cd treatment as compared to the same strains cultured in normal conditions (0 mg/L Cd). (**c**) Heatmap of parts of common genes for C1 and J45 strains after Cd treatment. (**d**) Heatmap of the common 14 gene for C78 and J127 strains after Cd treatment. The heat map showed a double hierarchical cluster among DEGs (vertical) and samples (horizontal). The different colors of the heatmap, ranging from blue over white to red, represent scaled expression levels of genes with (log2 (FPKM + 1)) across different samples.

**Figure 8 jof-08-00724-f008:**
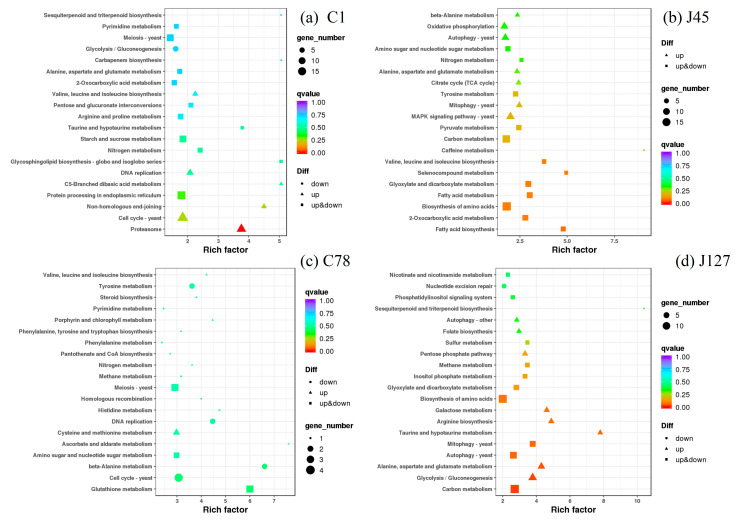
KEGG enrichment analysis of the DEGs in four strains (C1, J45, C78, and J127) of *C. geophilum* after 2 mg/L Cd treatment as compared to the same strains cultured in normal conditions (0 mg/L Cd). Each symbol (circle, square, and triangle) in the figure represents a KEGG pathway, the ordinate indicates the name of the pathway, and the abscissa is the enrichment factor. The size of the symbol represents the numbers of DEGs involved in the corresponding pathways.

**Figure 9 jof-08-00724-f009:**
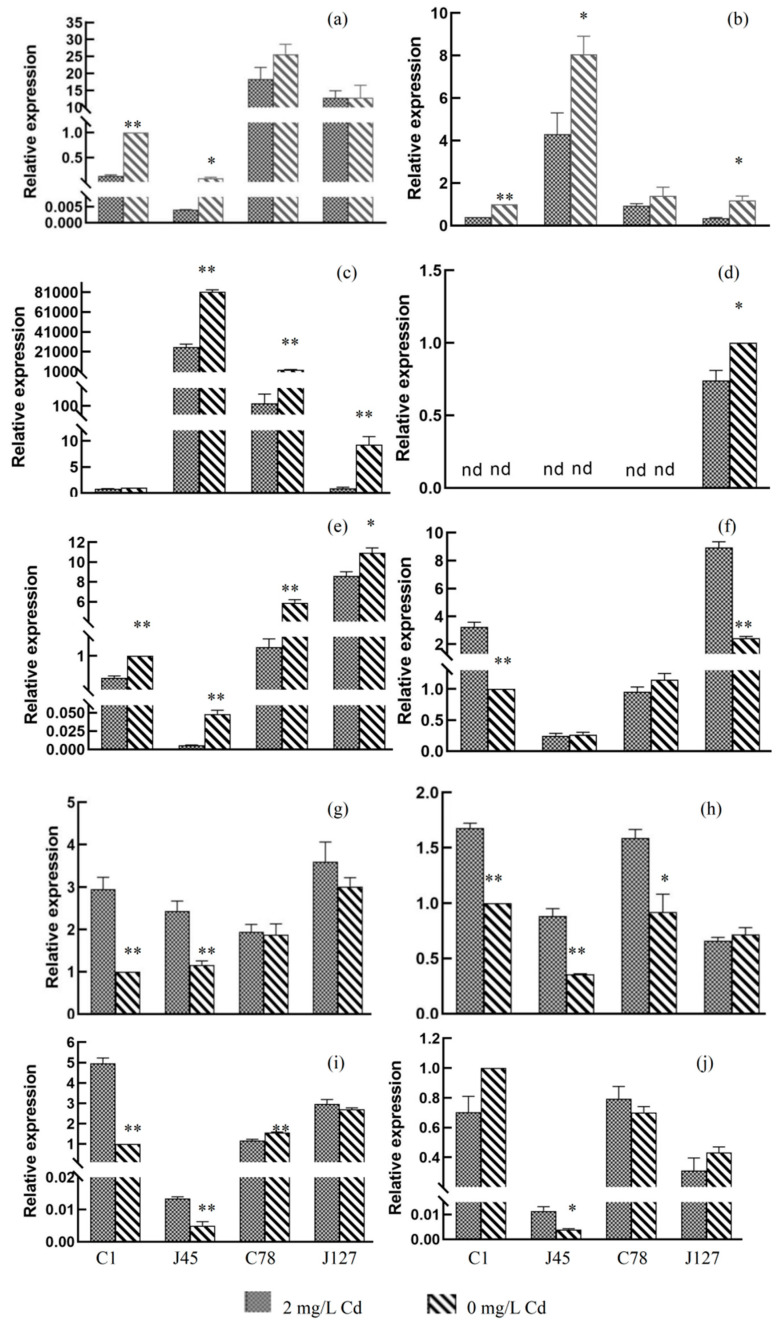
RT-qPCR analysis of the expression level of ten candidate genes in different samples. (**a**) gene-K441DRAFT-552345, (**b**) gene-K441DRAFT-600384, (**c**) gene-K441DRAFT-611738, (**d**) gene-K441DRAFT-660325, (**e**) gene-K441DRAFT-681328, (**f**) gene-K441DRAFT-653729, (**g**) gene-K441DRAFT-654069, (**h**) gene-K441DRAFT-668187, (**i**) gene-K441DRAFT-672999, and (**j**) gene-K441DRAFT-603024. * *p* < 0.05; ** *p* < 0.001. The relative expression changes of the target gene in the control group and the experimental group were calculated by 2^−ΔΔCt^ method, and shown as means (*n* = 3) ± SD.

## Data Availability

The data presented in this study are available within the article and Appendix A. *Cenococcum geophilum* raw sequence reads data available in a publicly repository [NCBI], reference number [PRJNA835631].

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
