# Peer review of "Comparative Physiological and Transcriptome Analysis Provide Insights into the Response of Cenococcum geophilum, an Ectomycorrhizal Fungus to Cadmium Stress"

_jof, 2022, doi:10.3390/jof8070724_

Round 1

Reviewer 1 Report

Comment 1: Line no. 20 Please avoid Cg for the scientific name, change in to C. geophilum. Please check throughout the manuscript and revise.

Comment 2: Line no. 24, expand DEG

Comment 3: Line no. 25-29, rewrite the sentences.

Comment 4: Please include some information about antioxidant enzymes in Introduction section.

Comment 5: Line no. 362, no need mention again full scientific name of an organism.

Comment 6: Line no. 365, give the number for citation.

Comment 7: Lot of information in the results, please improve the discussion section.

Comment 8: Rewrite the conclusion.

Author Response

Comment 1: Line no. 20 Please avoid Cg for the scientific name, change in to C. geophilum. Please check throughout the manuscript and revise.

Response 1: Thank you for your suggestion. We have changed Cg of the full text to C. geophilum, please check throughout the manuscript.

Comment 2: Line no. 24, expand DEG

Response 2: Yes, thank you for your suggestion. We have changed it, and wrote the full name of DEG (differentially expressed gene), please check in lines 25-26.

Comment 3: Line no. 25-29, rewrite the sentences.

Response 3: Thank you for your comments. We have rewritten the sentences on lines 25-29, the new sentence is “Comparative transcriptomic analysis was used to identify differentially expressed genes (DEGs) of four selected C. geophilum strains after 2 mg/L Cd treatment. The results showed that the defense response of C. geophilum strain to Cd may be closely related to the differential expression of functional genes involved in cell membrane ion transport, macromolecular compound metabolism and redox pathways. The results were further confirmed by RT-qPCR analysis. Collectively, this study provides useful information for elucidation of the Cd tolerance mechanism of ECMF.”, please check in lines 25-31 .

Comment 4: Please include some information about antioxidant enzymes in Introduction section.

Response 4: Thank you for your suggestion. We have added some information about antioxidant enzymes in the introductory section, and confirmed them with citations, please check in Lines 72-76.

Comment 5: Line no. 362, no need mention again full scientific name of an organism.

Response 5: Thank you for your comments. We have checked the scientific name of organism of the full manuscript and changed in to its abbreviation, please check in Lines 209, 226-227, 247, 298, 312-313, 327, 354-355.

Comment 6: Line no. 365, give the number for citation.

Response 6: Thank you for your suggestion. We have checked it. The content of line 365 in the text describes the content of our research results in figure 9. The methods for calculating relative expression changes are supplemented in materials and methods, please check in Line 190.

Comment 7: Lot of information in the results, please improve the discussion section.

Response 7: Thanks a lot for your comments. We have modified the discussion section; please see the details in Lines 381-501.

Comment 8: Rewrite the conclusion.

Response 8: Thank you for your suggestion. We have rewritten the conclusion. The revised conclusion content is “In this study, three Cd sensitive and three Cd tolerant strains of C. geophilum were selected as materials, and the responses to Cd were analyzed from the aspects of growth, enzyme activity, element absorption and gene expression. The results showed that under Cd stress, C. geophilum could resist the toxicity of Cd to the strains by increasing the antioxidant enzyme activity and nutrient absorption, reducing the absorption of Cd, and regulating the related genes encoding metabolism of compounds, ion transport over the cell membrane and antioxidant enzyme activity. In summary, these results provide some new insights into the physiological and molecular responses of C. geophilum to Cd treatment, which may help researchers understand the Cd stress responses of ECMF.”, please check in Lines 503-512.

Reviewer 2 Report

The authors rise up an interesting side related to ectomycorrhizal fungi Cd-tolerance. They based their study on an evaluation of the main physiological responses of Cenococcum geophilum (Cg) tolerant and sensitive strains and provided a transcriptomic analysis of Differentially expressed genes (DEGs) in mycelia of the cadmium treatment and control.

Over all, the results showed growth inhibition of Cg tested strains. Cd-tolerant Cenococcum geophilum strains exhibited lower Cd content and high antioxidant enzymes activities; which showed the less toxic effect of Cd on the tolerant Cenococcum geophilum strains. The molecular analysis indicated different genes involved in several mechanisms (ion transport, redox pathway, …) related to Cd exposure.

The manuscript is well drafted, structured and respecting the main guidelines of “the Instructions to authors”. The study fits with the aims of the journal. Recommendation to be considered in order to improve the manuscript:

- Line 24: Define the DEGs abbreviation on the first citation

- Line 112: The choice of Cd concentrations (0.25 and 2 mg/L) used for the evaluation of mycelial growth was based on what?? Clarify??

- Line 121: For the evaluation of the enzyme activities; only the highest Cd concentration (2 mg/L) was tested??

- For Line 138 (two Cg strains) and Line 147 (Four Cg strains) were choosen to evaluate respectively the Cd distribution in Cg strains mycelia and the RNA-seq analysis. Why different total strains (2 and 4) are considered??

- Results section: The titles of the provided figures essentially Figures: 1,2,3 and 4, must be rectified. In fact, Figure 1 (Line 198-202), Figure 2 (Line 214-217), Figure 3 (Line 232-236) and Figure 4 (Line 258-263) must be included in Material and methods section. Keep only indication of the presented results and Data analysis/ statistics and replication number in the figures titles.

Author Response

- Line 24: Define the DEGs abbreviation on the first citation

Response 1: Thanks a lot for your comments. We have redefined the DEGs abbreviation on the first citation, and wrote the full name of differentially expressed genes (DEGs). Please check in Lines 25-26.

- Line 112: The choice of Cd concentrations (0.25 and 2 mg/L) used for the evaluation of mycelial growth was based on what?? Clarify??

Response 2: Thanks a lot for your comments. The selection of cadmium (Cd) concentrations (0.25 and 2 mg/L) was used to evaluate the growth of Cenococcum geophilum (C. geophilum). Based on our screening experiments with different Cd concentrations, when the Cd concentration exceeded 2 mg/L, the growth of C. geophilum was severely inhibited. And it caused physiology toxicity to C. geophilum. Thus, low (0.25 mg/L) and high (2 mg/L) concentrations of Cd were finally selected to study the growth of different C. geophilum under Cd stress.

- Line 121: For the evaluation of the enzyme activities; only the highest Cd concentration (2 mg/L) was tested??

Response 3: Thanks a lot for your comments. From our previous selection experiments, it was found that the 2 mg/L Cd concentration was the maximum inhibition concentration of C. geophilum. Thus, we chose this concentration to study the resistance mechanism of C. geophilum under Cd stress.

- For Line 138 (two Cg strains) and Line 147 (Four Cg strains) were choosen to evaluate respectively the Cd distribution in Cg strains mycelia and the RNA-seq analysis. Why different total strains (2 and 4) are considered??

Response 4: Thanks a lot for your comments. In many studies, only representative strains were selected to analyze the distribution of heavy metals in mycelia. Therefore, we selected one sensitive and one tolerant strain for transmission electron microscope analysis.

- Results section: The titles of the provided figures essentially Figures: 1,2,3 and 4, must be rectified. In fact, Figure 1 (Line 198-202), Figure 2 (Line 214-217), Figure 3 (Line 232-236) and Figure 4 (Line 258-263) must be included in Material and methods section. Keep only indication of the presented results and Data analysis/ statistics and replication number in the figures titles.

Response 5: Yes, thank you for your comments. We have checked it, Figure 1 (Line 198-202), Figure 2 (Line 214-217), Figure 3 (Line 232-236) and Figure 4 (Line 258-263) have included in material and methods section. Please see the details in Lines 113-155, 209-213, 227-230, 247-251 and 273-277.
